# Dual Cluster Model for Medium-Range Order in Metallic Glasses

Masato Shimono [1,*] and Hidehiro Onodera [2]

1 National Institute for Materials Science, 1-2-1 Sengen, Ibaraki, Tsukuba 305-0047, Japan
2 Materials Design Technology Co., Ltd., 2-15-1 Konan, Minato, Tokyo 108-6028, Japan; amx17243@mail2.accsnet.ne.jp
* Correspondence: shimono.masato@nims.go.jp

**Abstract:** The atomic structure of medium-range order in metallic glasses is investigated by using molecular dynamics (MD) simulations. Glass formation processes were simulated by rapid cooling from liquid phases of a model binary alloy system of different-sized elements. Two types of short-range order of atomic clusters with the five-fold symmetry are found in glassy phases: icosahedral clusters (I-clusters) formed around the smaller-sized atoms and Frank–Kasper clusters (i.e., Z14, Z15, and Z16 clusters (Z-clusters)) formed around the bigger-sized atoms. Both types of clusters (I-and Z-clusters) are observed even in liquid phases and the population of them goes up as the temperature goes down. A considerable atomic size difference between alloying elements would enhance the formation of both the I- and Z-clusters. In glassy phases, the I- and Z-clusters are mutually connected to form a complicated network, and the network structure becomes denser as the structural relaxation goes on. In the network, the medium-range order is mainly constructed by the volume sharing type connection between I- and Z-clusters. Following Nelson's disclination theory, the network structure can be understood as a random network of Z-clusters, which is complimentarily surrounded by another type of network formed by I-clusters.

**Keywords:** metallic glasses; molecular dynamics; icosahedral symmetry; medium-range order; Frank–Kasper clusters; disclination; dense random packing; continuous random network





## 1. Introduction

The atomic-level structure of liquids and glasses is a long-standing problem in materials science. The dense random packing (DRP) model, originally proposed for liquids [1] and later applied to a structure of amorphous metals [2], indicates that the icosahedral cluster should be a key building block. The early simulation studies [3,4] have shown that the icosahedral order would exist in both liquid and glassy phases. After finding metallic glasses [5,6], experimental observations [7–12] have shown that the icosahedral short-range order does exist in glassy alloys and that some medium-range order may also exist beyond the icosahedral short-range order. Being inspired by two pioneering models [13,14] for a icosahedral medium-range structure, a family of network-type models has been proposed [15–21]. However, the topological feature of the icosahedral network is not clearly understood yet. To tackle this problem, Cheng and Ma have offered [22] a more comprehensive idea that the icosahedral order can be naturally understood if other types of the Frank–Kasper clusters [23] are included as building blocks in addition to the icosahedral cluster. This viewpoint is originated from the "disclination" theory for liquids and glasses proposed by Nelson [24], in which various types of the Frank–Kasper clusters are considered to evaluate the frustration energy in the DRP structure. Along this storyline, we think we should not only consider the icosahedral cluster but also other types of the Frank–Kasper clusters to understand the medium-range structure in metallic glasses. Hence, in the present study, we investigate structural properties of the icosahedral

medium-range order formed in metallic glasses with special attention to the Frank–Kasper clusters as well as the icosahedral cluster by using molecular dynamics (MD) simulations.

MD simulation is a powerful tool to investigate the atomic-scale structure because all information of atomic configurations can be drawn at any time in the course of calculations. The aim of our study is to clarify the topological feature of the icosahedral medium-range order in metallic glasses from the atomistic point of view. For this purpose, the MD technique is highly useful. This article is planned as follows. The methods of MD simulation are given in Section 2. The simulation results are shown in Section 3, where the glass-formation dynamics and the structural properties of glassy phases are investigated with paying special attention to the formation and percolation of the Frank–Kasper clusters. In Section 4, the geometrical and topological property of the network formed by Frank–Kasper clusters is discussed based on Nelson's disclination theory [24]. The conclusion is given in Section 5.

## 2. Methods

### 2.1. Interatomic Potentials

It is well known that the atomic size ratio between the alloying elements plays an important role in the formation of metallic glasses [25]. Therefore, as a simple model for binary alloys, we assume the interaction energy between atoms separated by the distance $r$ to be described by the Lennard–Jones (LJ)-type potential [26] $V^{ij}$ as

$$V^{ij} = e^{ij} \{(r^{ij}/r)^8 - 2(r^{ij}/r)^4 \}, \tag{1}$$

where $i$ and $j$ denote the atomic species and the parameters $r^{ij}$ and $e^{ij}$ correspond to the atomic size and the chemical bond strength, respectively. In this study, to focus on the atomic size effect, we assume for a binary system composed of elements A and B as $r^{AA} = 1$, $r^{BB} \leq 1$, $r^{AB} = (r^{AA} + r^{BB})/2$, and $e^{AA} = e^{BB} = e^{AB} = 1$. Thus, we can vary the atomic size ratio $r^{BB}$ of the element B to A, and the concentration $x$ of the smaller element B. The atomic masses of both elements are supposed to be the same unit mass $m_A = m_B = 1$. In this paper, all physical quantities are expressed in the above LJ units, that is, the lengths and volumes are expressed by the unit $r^{AA} = 1$, the energies and temperatures are expressed by the unit $e^{AA} = 1$, the masses are expressed by the unit $m_A = 1$, and the time intervals and rates are expressed by the unit $(m_A/e^{AA})^{1/2} r^{AA} = 1$.

### 2.2. Simulation Procedure

The simulation system consists of 16,000 atoms. All atoms are confined in a cubic box, in which periodic boundary conditions are imposed along all three directions. The temperature of the system is controlled by scaling the atomic momenta. The pressure of the system is kept zero by changing the size of the simulation cell according to the constant pressure formalism [27].

In the simulation, an A-B model alloy system starts from a liquid state annealed at above the melting point and then cooled down to solidify. The quenching processes are performed by three different cooling rates: $2 \times 10^{-4}$, $2 \times 10^{-5}$, and $2 \times 10^{-6}$, which we call fast, middle-, and slow-cooling, respectively. By monitoring the volume, energy, radial distribution of atoms, and the atomic mobility, we can detect the phase transition from liquid phases into glassy or crystalline phases.

### 2.3. Voronoi Tessellation and Frank–Kasper Clusters

To investigate the local atomic structure of liquid and glassy phases with paying a special attention to the icosahedral symmetry, we use the Voronoi tessellation analysis [1]. In the Voronoi analysis, the local symmetry around each atom is indexed by a set of integers $(n_3, n_4, n_5, n_6)$, where $n_i$ is the number of $i$-edged faces of the Voronoi cell, which is defined by the polygon surrounded by the bisecting planes between the corresponding atom and its neighboring atoms. By using the Voronoi index, the center atom of the icosahedral cluster is indexed as (0 0 12 0).

As a structural unit, not only the icosahedral clusters indexed as (0 0 12 0) but other types of clusters also play an important role in liquid and glassy phases of metallic glasses. Recently, in the simulation study, Ding et al. has shown [28] that icosahedral clusters are only dominated in the Cu-centered clusters, but another type of Frank–Kasper cluster indexed as (0 0 12 4), called the "Z16 cluster", is dominated in the Zr-centered clusters and both types of clusters contribute to the icosahedral order formation in the $Cu_{64}Zr_{36}$ supercooled liquids. Therefore, we should pay special attention to the Frank–Kasper clusters with neighbors more than twelve, as well as the icosahedral cluster. As will be shown in the next section, in addition to the icosahedral cluster and the Z16 cluster, we can find several types of Frank–Kasper clusters in glassy phases of the model A-B system, that is, icosahedron or Z12 indexed as (0 0 12 0), Z14 indexed as (0 0 12 2), Z15 indexed as (0 0 12 3), and Z16 indexed as (0 0 12 4), as shown in Figure 1. As denoted by the red marks in Figure 1, the Z14, Z15, and Z16 clusters have two, three, and four hexagonal Voronoi faces, respectively, as well as twelve pentagonal Voronoi faces. Hereafter, we call Z12 as I-cluster and Z14, Z15, and Z16 as Z-clusters (a little different definition found in Ref. [22]).

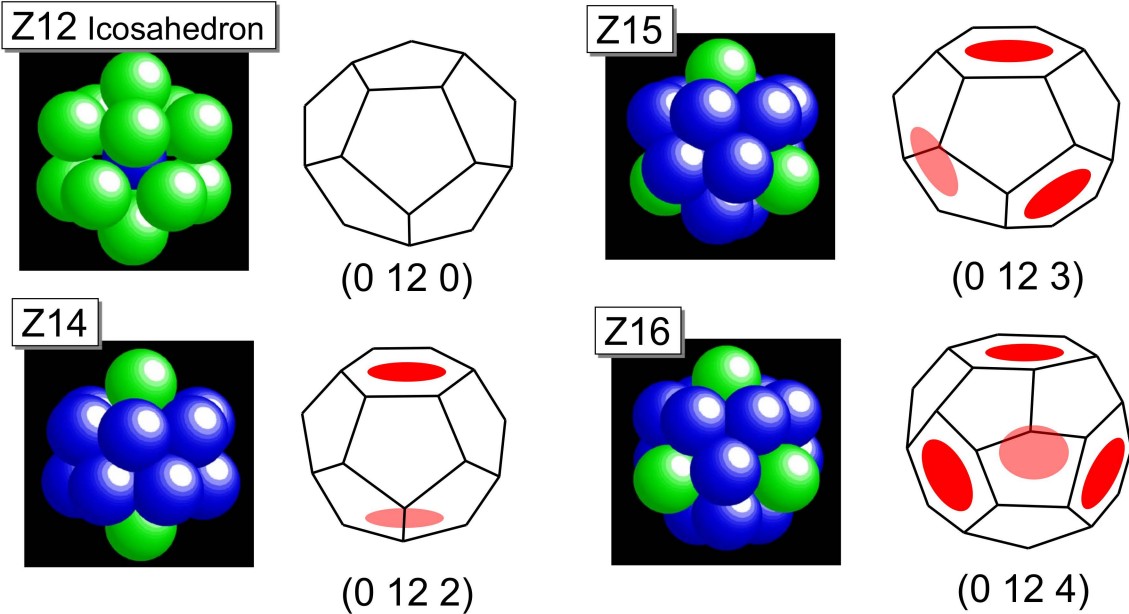

**Figure 1.** Four different types of Frank–Kasper clusters: Z12, Z14, Z15, and Z16, and their Voronoi polyhedra found in the glassy A-B phases. The green and blue spheres denote the A and B atoms, respectively. The six-edged Voronoi faces (hexagons) are indicated by the red marks.

## 3. Results

### 3.1. Glass Formation by Liquid Quenching

The glass-forming range by rapid quenching from liquid phases in the model A-B binary system used here has been investigated in our previous studies [29,30]. In the A-B binary system, the good glass-former satisfies one of Inoue's empirical rules [25], that is, a large atomic size difference more than 10%. Figure 2a shows the evolution of the atomic volume during a middle-cooling process (cooling rate $2 \times 10^{-5}$) of the $r^{BB} = 0.9$ $A_{50}B_{50}$ system. The change in the slope of the volume dependence corresponds to the glass transition point $T_g$. During the cooling process, the radial distribution function (RDF) is changed, as shown in Figure 2b, where the splitting of the second peak, which is known to be a sign of glass formation, can be clearly observed below $T_g$ ($T = 0.1$) and slightly observed near $T_g$ ($T = 0.3$). As well as the RDF, the Voronoi analysis indicates no sign of crystallization in the solidified phase at $T = 0.001$, whose atomic configuration is shown in Figure 2c.

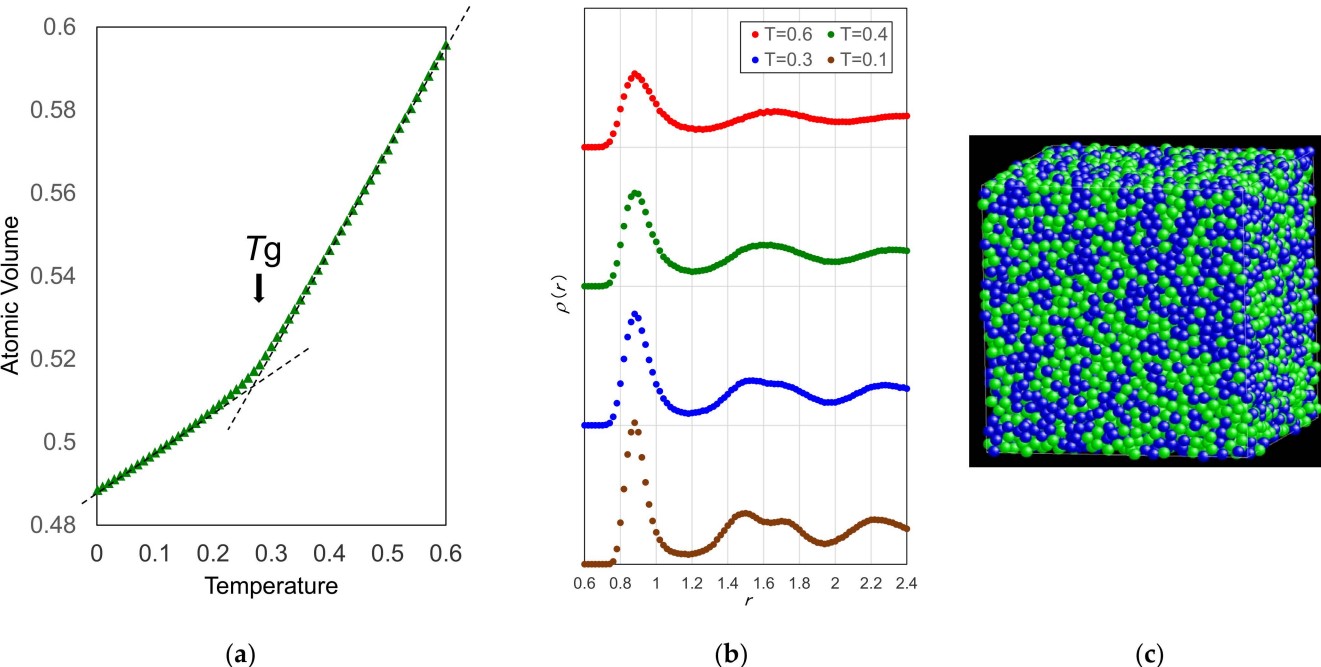

**Figure 2.** (**a**) Temperature dependence of the atomic volume in a middle-cooling process of the $r^{BB}$ = 0.9 $A_{50}B_{50}$ system. (**b**) Radial distribution functions at $T$ = 0.6, 0.4, 0.3, and 0.1. (**c**) Atomic configuration of an as-quenched state at $T$ = 0.001, where the green and blue spheres denote the A and B atoms, respectively.

To explore the icosahedral or five-fold symmetry grown in liquid and glassy phases, we next estimate the population of the I- and Z-clusters shown in Figure 1 during cooling processes from liquids. In Figure 3a, the evolution of the population of the I- and Z-clusters in a middle-cooling process of the $r^{BB}$ = 0.8 $A_{50}B_{50}$ system are shown. The formation of I- and Z-clusters is observed even in liquid phases, and both of the I- and Z-clusters are rapidly growing in the supercooled regime when the temperature approaches the glass transition point. The whole system would be covered by these clusters at around $T_g$, as shown in the upper inset of Figure 3a.

The atomic mobility rapidly decreases near $T_g$ and is believed to induce the 'structural freezing' of the system. Figure 3b shows the time evolution of mean square displacements of the A atoms (green), B atoms (blue), the central atoms of I-clusters (red), and those of Z-clusters (yellow) in an $r^{BB}$ = 0.8 $A_{50}B_{50}$ supercooled liquid phase annealed at $T$ = 0.36. We can estimate the atomic mobility from the slopes depicted in Figure 3b. The results of the fitted values are 0.0083, 0.0113, 0.0005, and 0.0009 for the A atoms, B atoms, the central atoms of I-clusters, and those of Z-clusters, respectively. In average, the B atom shows 30% higher mobility than the A atom due to its smaller size. In supercooled liquids, I- and Z-clusters are relatively more stable than others [28,31] but almost all clusters decay in 10,000 MD steps, which corresponds 50 in the LJ unit. So, the average values for the I- and Z-clusters in Figure 3b are taken from only 138, 5, 17, and 18 events for the I-, Z14, Z15, and Z16 cluster, respectively. The events for the Z14 cluster are relatively few due to it having less stability comparing to Z15 and Z16 clusters in supercooled liquids. In spite of the poor statistics, it is clear that the atomic mobility would become more than 10 times lower if the atoms would form an I- or Z-cluster in the supercooled liquid phase, which may induce the structural freezing near the glass transition temperature.

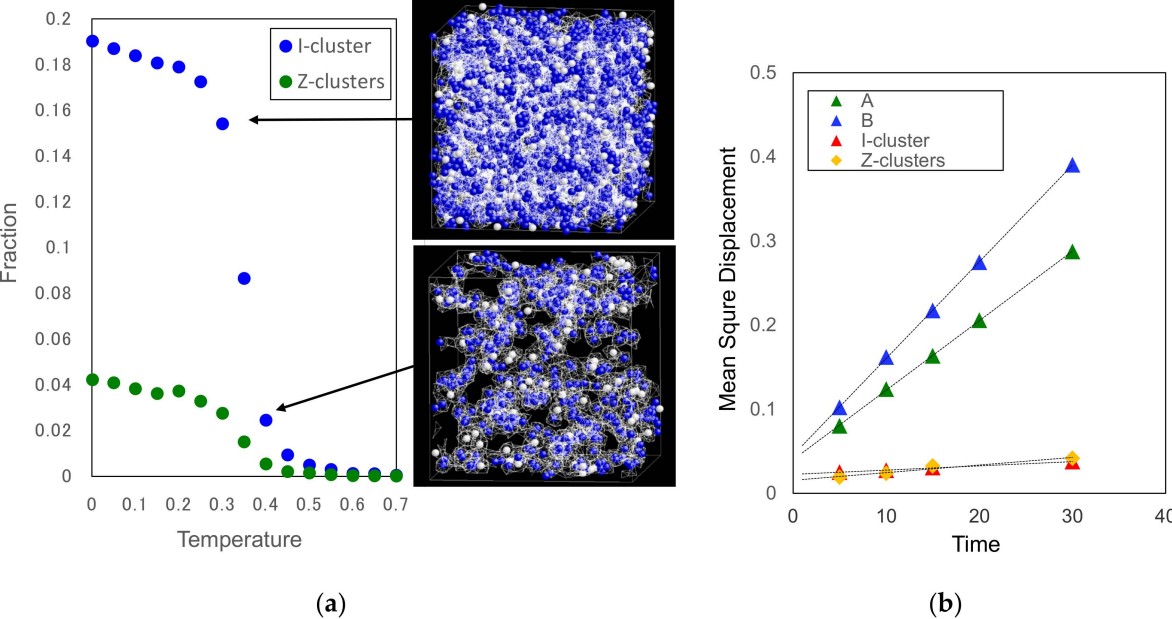

**Figure 3.** (**a**) Temperature dependence of the population of I- and Z-clusters in a middle-cooling process of the $r^{BB} = 0.8$ $A_{50}B_{50}$ alloy system. The insets show the spatial distribution of I- and Z-clusters at $T = 0.3$ (upper) and $T = 0.4$ (lower), where the central atoms of the I- and Z-clusters are depicted by the blue and white spheres, respectively. (**b**) Time evolution of mean square displacements of the A atoms (green), B atoms (blue), the central atoms of I-clusters (red), and those of Z-clusters (yellow) in a supercooled liquid phase of the same system annealed at $T = 0.36$.

### 3.2. Icosahedral Order in Glassy Phases

#### 3.2.1. Cooling Rate Dependence of Icosahedral Order

If we obtain glassy phases in the same alloy system with the different cooling rate, we can investigate the effect of structural relaxation in glassy states. For the $r^{BB} = 0.8$ $A_{50}B_{50}$ system, the evolution of the atomic energy in the quenching processes with different cooling rates ranging from $2 \times 10^{-4}$ to $2 \times 10^{-6}$ is shown in Figure 4a. The solidified phase shows lower energy for a lower cooling rate, which indicates that more structural relaxation would occur in the glassy phase formed by lower cooling rate. For each solidified phase at $T = 0.001$, the fraction of the I- and Z-clusters was calculated, and the results are shown in Figure 4b. As well as the I-cluster, the Z-clusters also increase as the cooling rate decreases. Since the lower cooling rate would bring the more relaxed glassy structure, it indicates that both the I- and Z-clusters should be essential building blocks in the glassy phases. We also showed the fraction of the atom species (A or B) of the central atoms of the I- and Z-clusters in the glassy $A_{50}B_{50}$ phase formed by slow-cooling (cooling rate $2 \times 10^{-6}$) in Figure 4c. As expected [28], 98% of I-clusters are centered by the (smaller) B atoms, while 95% of the Z-clusters are centered by the (bigger) A atoms.

#### 3.2.2. Atomic Size Effect on Icosahedral Order

Atomic size difference between alloying elements plays a decisive role in glass-forming ability of alloy systems [25]. We calculated the dependence of the population of I- and Z-clusters on the atomic size ratio $r^{BB}$ in the glassy phases of the $A_{50}B_{50}$ system formed by slow-cooling processes. The results are shown in Figure 5a. The population of the both I- and Z-clusters increase as the atomic size difference increases up to 0.2 ($r^{BB} = 0.8$), while they turn to decrease beyond a 20% atomic size difference. Note that the atomic size difference of 0.2 approximately corresponds to the Zr–Cu system, which is known as a prototype of binary good glass-formers.

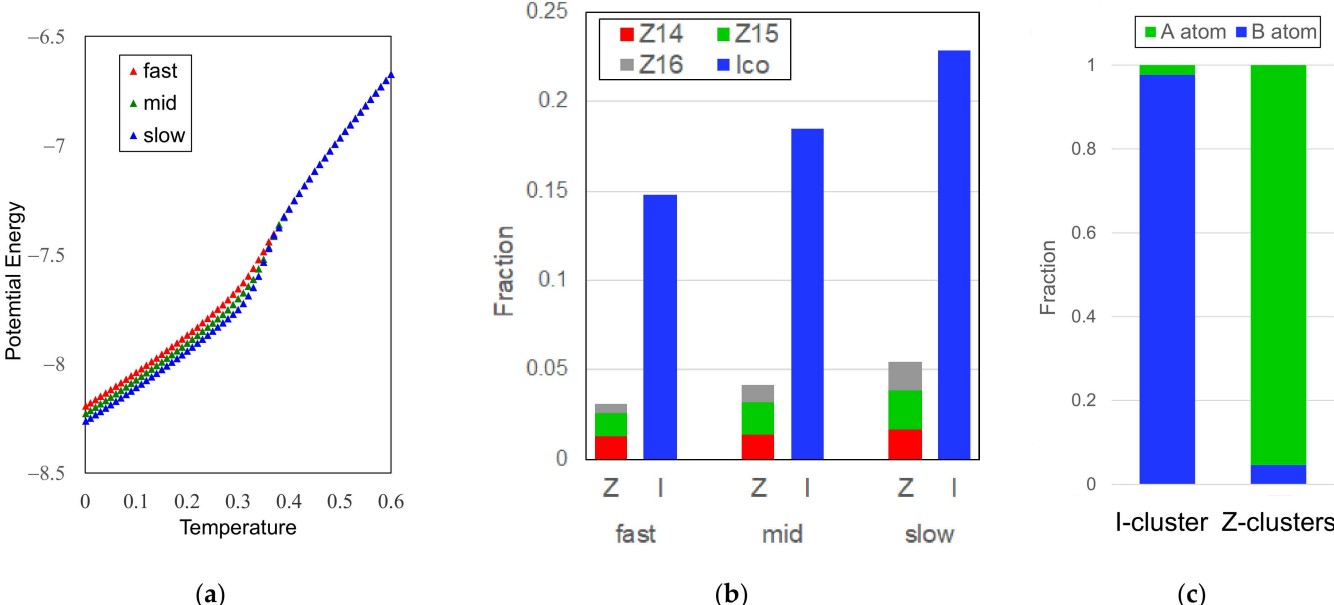

(a)  (b)  (c)

**Figure 4.** (**a**) Temperature dependence of potential energy in cooling processes of the $^{r\mathrm{BB}}$ = 0.8 $A_{50}B_{50}$ system with different cooling rates. (**b**) Cooling rate dependence of the fraction of I- and Z-clusters in quenched glassy $A_{50}B_{50}$ phases. (**c**) Fraction of atom species of the central atoms of I- and Z-clusters in the glassy $A_{50}B_{50}$ phase formed by slow-cooling.

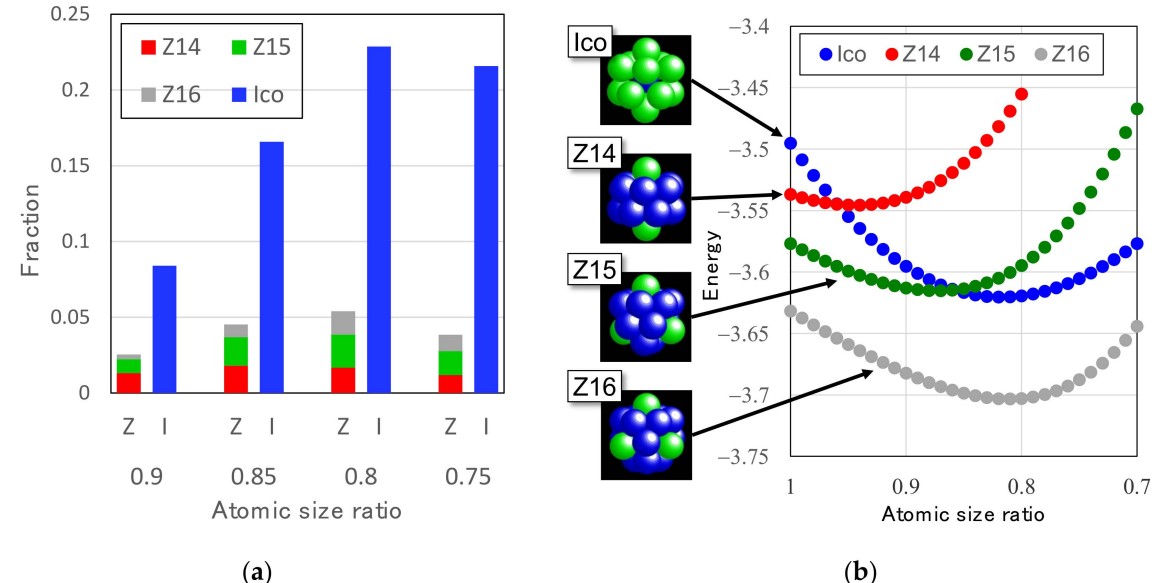

(a)  (b)

**Figure 5.** (**a**) Atomic size dependence of the population of the fraction of I- and Z-clusters in quenched glassy $A_{50}B_{50}$ phases formed by slow-cooling processes. (**b**) Atomic size dependence of the atomic energy of I- and Z-clusters. Atomic configuration of each cluster is shown in the insets, where the green and blue sphere denote the A and B atoms, respectively.

To check the relation between the cluster stability and the atomic size ratio, we calculated the dependence of cluster energy per atom on the atomic size ratio. The results are shown in Figure 5b. As shown in the insets of Figure 5b, we fixed the atomic configuration of each Frank–Kasper cluster from a geometrical point of view. For the I-cluster, the central atom is a (smaller) B atom surrounded by twelve (bigger) A atoms. For Z14, Z15, and Z16 clusters, the central atom and the neighboring atoms sharing a hexagonal face with the central atom are (bigger) A atoms and the rest twelve neighboring atoms are (smaller) B atoms. The atomic size ratios which correspond to the minimum energy are 0.82, 0.94, 0.87,

and 0.81 for the I-, Z14, Z15, and Z16 cluster, respectively. Although we should consider the other types of atomic configuration of clusters for more correct evaluation of the cluster stability, we believe that the dependence shown in Figure 5b indicates that the glass-forming ability and the local icosahedral symmetry would be enhanced by introducing a large atomic size difference beyond 10%.

### 3.2.3. Concentration Dependence of Icosahedral Order

To investigate the concentration dependence of the icosahedral order in the $A_{1-x}B_x$ glassy phases, we calculated the population of I- and Z-clusters in the glassy phases of the $A_{1-x}B_x$ system formed by middle-cooling processes with varying the concentration $x$ of B and with fixing the atomic size ratio as $r^{BB} = 0.8$. The results are shown in Figure 6. It indicates that the icosahedral symmetry and the glass-forming ability are high in the concentration range $x = 0.55$–$0.70$, which agrees well with the simulation results of the previous studies [29,30].

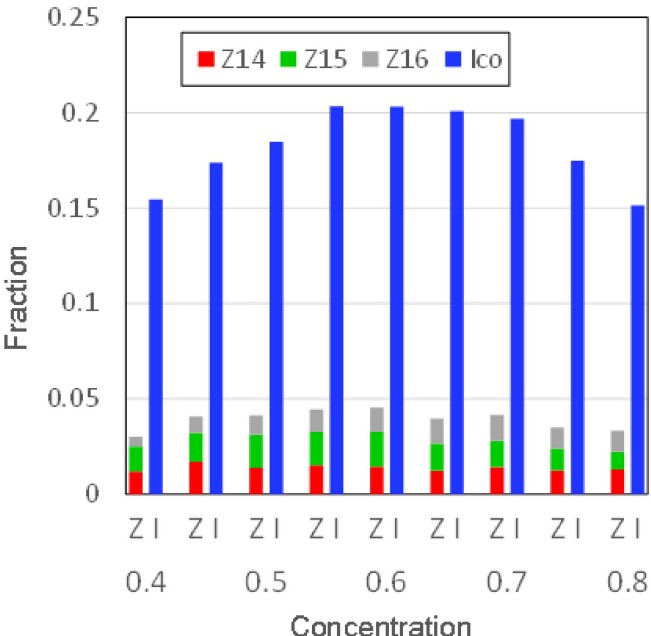

**Figure 6.** Concentration dependence of the population of the I- and Z-clusters in the glassy phases of the $r^{BB} = 0.8$ $A_{1-x}B_x$ system formed by middle-cooling processes.

### 3.3. Topological Feature of Icosahedral Medium-Range Order

#### 3.3.1. Linking Patterns between F-K Clusters

As shown in Figure 3a, a lot of I- and Z-clusters are formed even in supercooled liquid phases and their population goes up to form a complicated network in glassy phases. To investigate the topological feature of the network, we first focus on the linking pattern between I- and Z-clusters. When two I- or Z-clusters are linked together, the linking patterns can be classified into the following four types [15], as illustrated in the insets of Figure 7a: (1) vertex sharing, where one atom is shared by two clusters; (2) edge sharing, where two atoms forming a link are shared; (3) face sharing, where three atoms which form a triangle are shared; and (4) bicap sharing, where seven atoms which form a pentagonal bicap (bipyramid) or eight atoms which form a hexagonal bicap are shared or two clusters interpenetrate each other. We calculated the population of these four linking patterns for the $r^{BB} = 0.8$ $A_{50}B_{50}$ glassy phases formed with different cooling rates. The results are shown in Figure 7, where the population of the four connection types calculated individually between I-clusters (Figure 7a), between Z-clusters (Figure 7b), and between I- and Z-clusters (Figure 7c). In each type of connection, between I's, or between Z's, or between I and Z,

the dominant connecting pattern is the bicap-sharing- or interpenetrating-type, and the population of this type of connection increases as the structural relaxation takes place in glassy phases. Therefore, we believe that the bicap-sharing connection is the basic linking pattern in the network formed by I- and Z-clusters in glassy phases. This is consistent with recent experimental observations [12] using scanning electron nanodiffraction which suggests a face-sharing or bicap-sharing model of the icosahedral medium-range order in a $Zr_{36}Cu_{64}$ glass.

In their simulation study on the formation of amorphous iron, Pan et al. reported [32] that the ratio between different linking patterns between clusters would be changed during the liquid-to-amorphous transition. In liquid phases, the population of an edge-sharing connection is larger than that of a face-sharing connection. As the temperature decreases, the face-sharing connections rapidly grow and the population of the face-sharing connection becomes larger than that of edge-sharing connections in amorphous phases. So, we also calculated the temperature dependence of linking patterns for the connections between I-clusters in a middle-cooling process for the $r^{BB} = 0.8$ $A_{50}B_{50}$ system. The results are shown in Figure 7d. In this case, the ratios between four different types are almost unchanged during cooling, which indicates that the bicap-sharing connection would be the most basic pattern in supercooled liquid phases as well as in glassy phases of the model alloy system.

Figure 8 shows the whole network structure formed by I- and Z-clusters connecting via bicap sharing in an $r^{BB} = 0.8$ $A_{40}B_{60}$ glassy phase formed by middle-cooling. Figure 8a shows all atoms belonging I- and Z-clusters in the network, where the green and blue spheres denote the A and B atoms, respectively, while Figure 8b shows only the central atoms of the I- and Z-clusters by the blue and white spheres, respectively, together with the bicap-sharing connections between them denoted by the sticks. Even after restricting the bicap-sharing connection, the structure looks so complicated that it would be a hard task to understand the topological feature of the network.

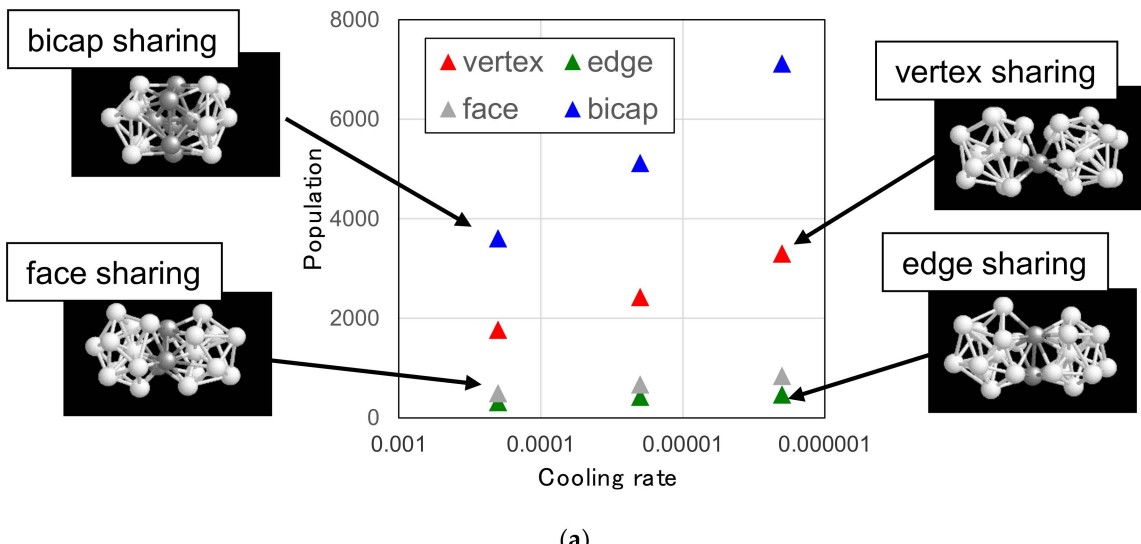

(**a**)

**Figure 7.** *Cont.*

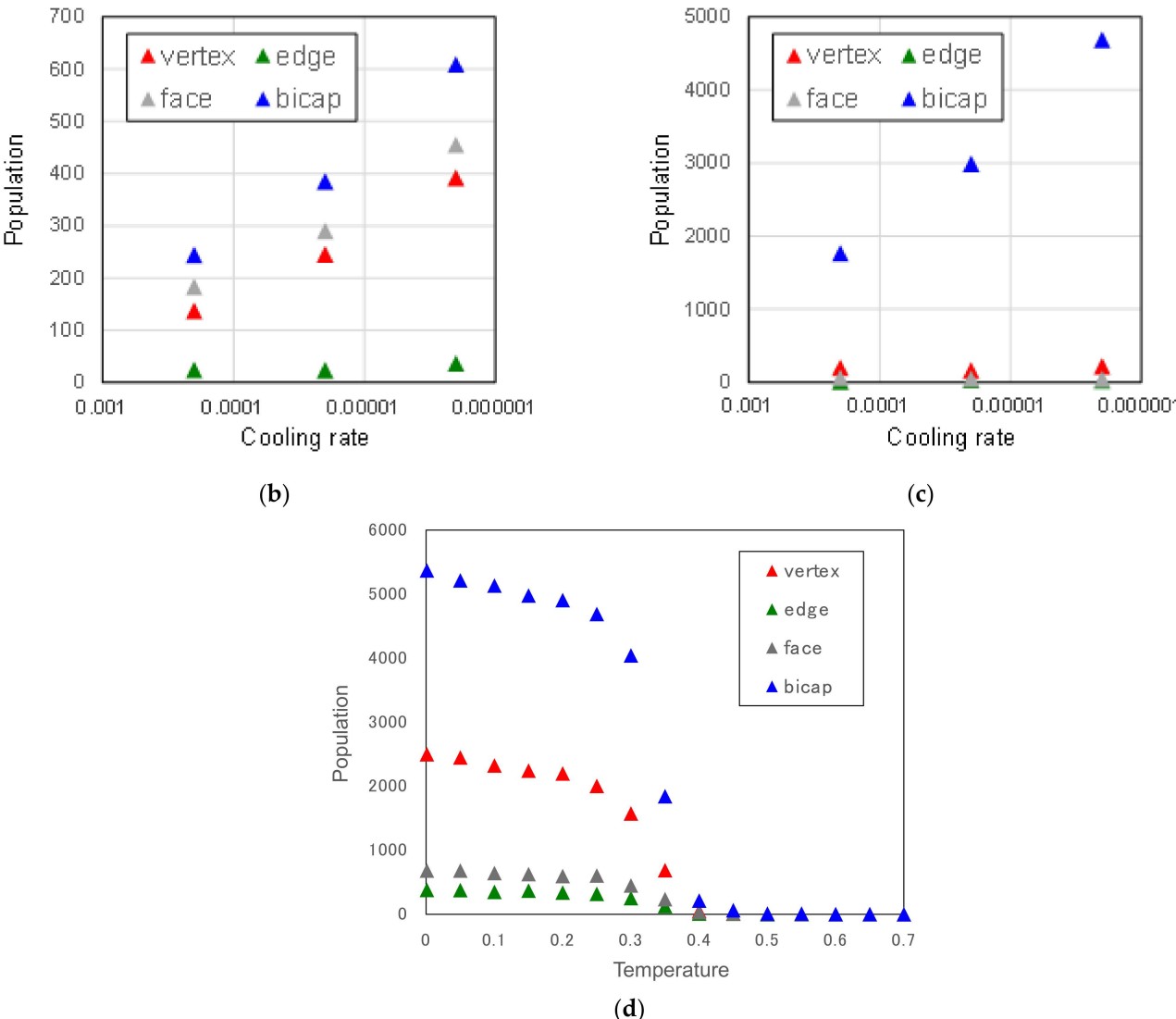

**Figure 7.** Possible linking patterns between I- or Z-clusters (insets), where the sharing atoms are denoted by grey spheres, and the dependence of their population in the $r^{BB}$ = 0.8 $A_{50}B_{50}$ glassy phase on the cooling rate: (**a**) Between I-clusters, (**b**) between Z-clusters, and (**c**) between I- and Z-clusters, and (**d**) temperature dependence of the population in a middle-cooling process for the $r^{BB}$ = 0.8 $A_{50}B_{50}$ system.

### 3.3.2. Typical Structural Unit in the Network

To understand the structural property of the network formed by I- and Z-clusters, we firstly focus on the network formed by I-clusters only. Figure 9a shows the network structure formed by I-clusters connecting via pentagonal bicap sharing in an $r^{BB}$ = 0.8 $A_{50}B_{50}$ glassy phase formed by slow-cooling. We can find a typical ring structure formed by connecting six I-clusters, as denoted by white circles. This structural unit is often observed in glassy phases given in simulation studies for model alloy systems [15], the TiAl system [19,33], and the CuZr system [22,34]. Among these studies, Xie et al. have pointed out [33] an important structural role: the stability and connectivity of the hexagonal unit in the icosahedral network. We calculated the population of the hexagonal ring unit in the $r^{BB}$ = 0.8 $A_{50}B_{50}$ glassy phases formed by different cooling rates. The results are shown in Figure 9b. The population goes up as the structural relaxation decreases. It indicates that the ring structure formed by six I-clusters would be a basic unit in the medium-range order in the network.

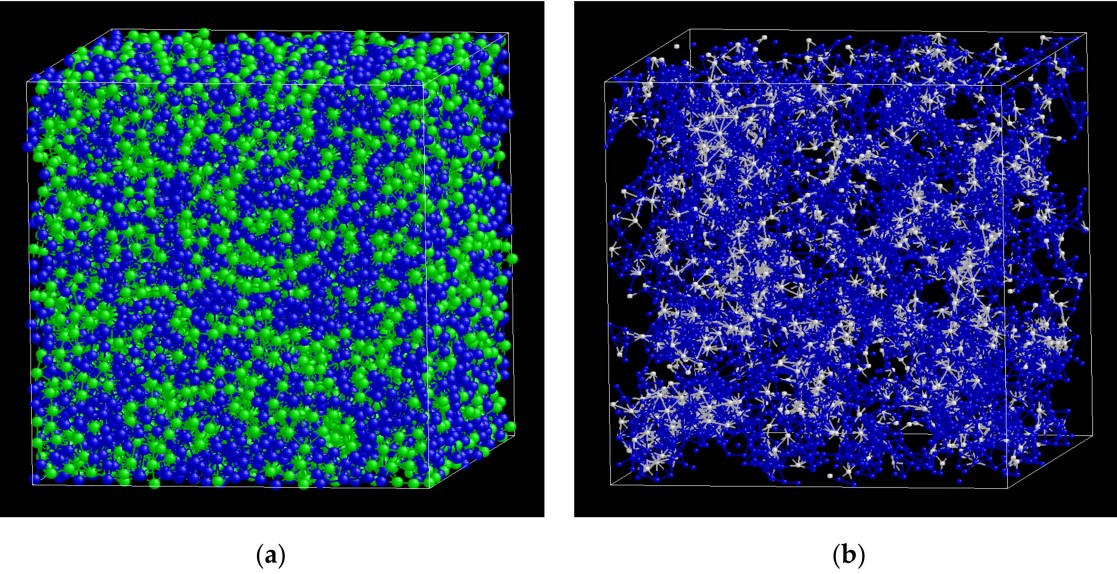

**Figure 8.** The network structure formed by I- and Z-clusters in an $r^{BB}$ = 0.8 $A_{40}B_{60}$ glassy phases formed by middle-cooling:
(**a**) All atoms belonging I- and Z-clusters in the network, where the green and blue spheres denote the A and B atoms, respectively. (**b**) The central atoms of I- and Z-clusters denoted by the blue and white spheres, respectively, together with the bicap-sharing connections between them.

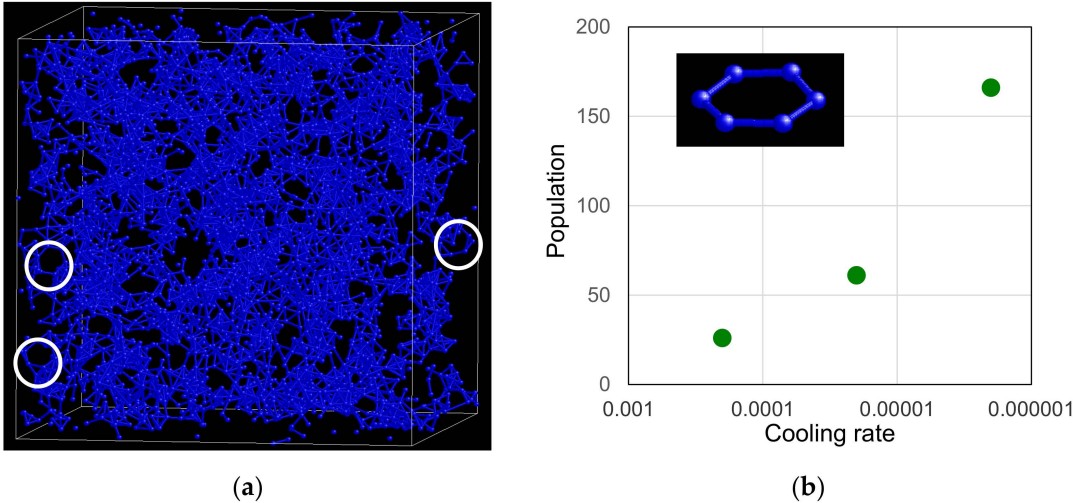

**Figure 9.** (**a**) The network structure formed by I-clusters in an $r^{BB}$ = 0.8 $A_{50}B_{50}$ glassy phase formed by middle cooling. (**b**) The cooling rate dependence of the population of the hexagonal ring structure (inset) formed by six I-clusters found in the bicap-sharing network in the $r^{BB}$ = 0.8 $A_{50}B_{50}$ glassy phases.

We next proceed to clarify the relation of the I-cluster network to the Z-cluster network by focusing on the hexagonal ring unit found above. By investigating the network structure around the hexagonal ring units with considering the bicap-sharing connection between Z's and between I and Z as well as the connection between I's, we found that the hexagonal unit formed by six I-clusters is often penetrated by a hexagonal bicap-sharing bond between two Z-clusters, as shown in Figure 10a, for example, at the portions denoted by the white circles. Thus, we propose a structural unit that extends further, as shown in Figure 10b (six pentagonal bicap-sharing bonds between six I-clusters and one hexagonal bicap-sharing bond between two Z-clusters), Figure 10c (all atoms belonging to the unit structure). Interestingly, this structure is the same as one of the structural units in "Frank–Kasper phases" or "topologically close pack phases", such as C14 and C15. Here, we come to an idea that some structural similarity in short-range order might exist between the metallic

glasses and the Frank–Kasper phases. Based on the idea, the individual roles of the I- and Z-clusters in forming the icosahedral medium-range order will be discussed in the next section.

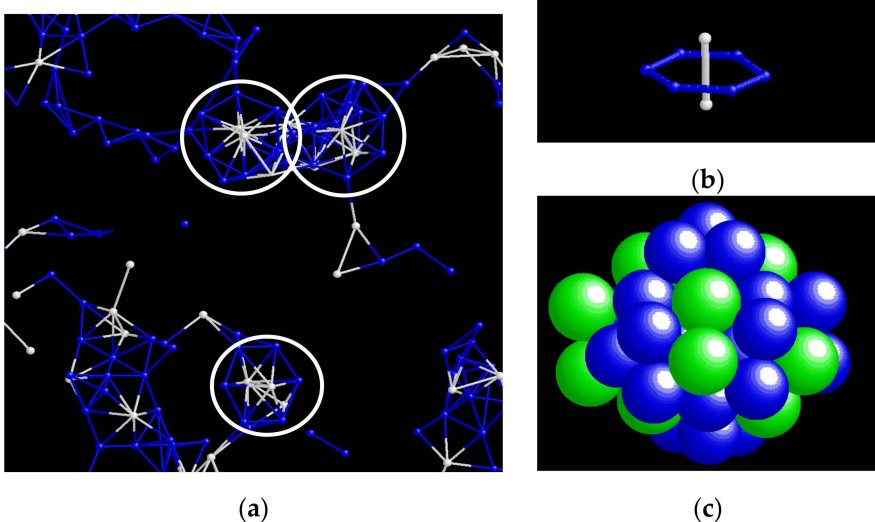

(b)

(a)                                                                  (c)

**Figure 10.** (**a**) A portion of network formed by I- and Z-clusters connecting via bicap-sharing found in a glassy phase of the $r^{BB} = 0.8$ $A_{50}B_{50}$ system, where spheres denote the central atoms of I-clusters (blue) and Z-clusters (white) and sticks denote the bicap-sharing bonds between them. Snapshots of atomic configuration of a typical unit found in the same $A_{50}B_{50}$ glassy phase: (**b**) Six pentagonal bicap sharing bonds (blue) between six I-clusters penetrated by a hexagonal bicap-sharing bond (white) between two Z-clusters. (**c**) All atoms belonging to the unit structure, where green and blue spheres denote the A and B atoms, respectively.

## 4. Discussion

### 4.1. Geometrical Features of Connection between I- and Z-Clusters

Depending on the shape of the corresponding Voronoi face, we can classify the bicap-sharing connections into two categories: one is a pentagonal bicap-sharing connection or the connection through a pentagonal face, and the other is a hexagonal bicap-sharing connection through a hexagonal face. As schematically shown in Figure 11, I-clusters have only pentagonal-type connections, while the Z-clusters have both pentagonal- and hexagonal-type connections. Consequently, the hexagonal-type connection should only exist between Z-clusters, because the I-cluster has no hexagonal faces. From this viewpoint, an interesting feature of the structural unit shown in Figure 10b is that the connection between two Z-clusters is a hexagonal bicap sharing or through a hexagonal face. In this sense, if we shall pick up only hexagonal-type connections between Z-clusters, we might understand the essential feature of the network formed by I- and Z-clusters by simplifying the topology of whole complicated structure. This viewpoint is the exact same as the "disclination" theory proposed by Nelson [24], which will be explained in the following subsections.

### 4.2. DRP Model and Regge Calculus

In three dimensions, the DRP structure is considered to be a space-filling with the tetrahedra. The fact that the regular tetrahedron has a dihedral angle of 70.5° which cannot completely fit to 360° is the reason why the DRP structure cannot fill the whole three dimensional space as crystalline structures do. Therefore, the DRP structure is always accompanied with frustration. To estimate this type of frustration or distortion energy, the Regge calculus [35] is an appropriate formalism, which was originally proposed as a model of theory of gravitation.



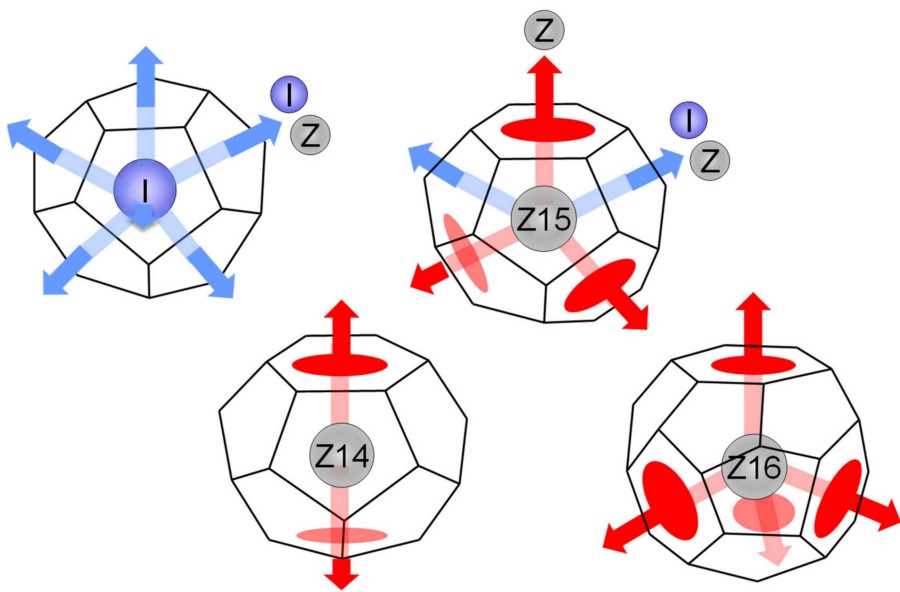

**Figure 11.** Schematics of the bicap-sharing connections between I- or Z-clusters through pentagonal faces (blue) and hexagonal faces (red).

In the Regge calculus, the distortion energy resides on the each link and can be estimated by the deficit angle, which is defined as the deviation from 360° of the sum of the dihedral angles of the tetrahedra surrounding the corresponding link. Depending on the number of sharing tetrahedra, we call the atomic bond as the 4-ring, 5-ring, or 6-ring bond, as illustrated in Figure 12. The deficit angles of the 4-, 5-, and 6-ring bond are calculated as 78°, 7°, and −63°, respectively. Consequently, the 5-ring bond has the lowest frustration or distortion, which is why the 5-ring bond or the five-fold topology dominates [3] in the DRP structure.

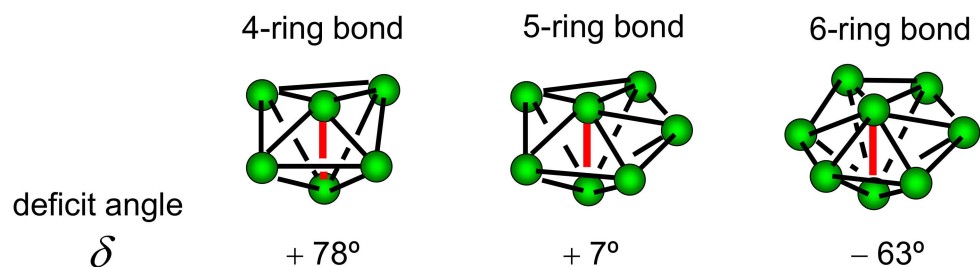

**Figure 12.** Schematics of the 4-, 5-, and 6-ring bonds and the deficit angle around each bond.

Based on the Regge calculus, the enhancement of the glass-forming ability by introducing a large atomic size difference between the alloying elements can be easily understood [31]. By introducing different sized elements, the variety of dihedral angles other than 70.5° arises, which allow for configurations of 5- or 6-ring bonds with lower frustration than those in monoatomic system. For example, in an A-B alloy system with the atomic ratio of 0.8, we can find the 5- and 6-ring bonds with lower frustration than half of those in monoatomic system, as shown in Figure 13. Because both I-clusters and Z-clusters have twelve 5-ring bonds and Z14, Z15, and Z16 clusters have additionally two, three, and four 6-ring bonds, respectively, the distortion energy of these clusters would also be reduced by introducing the atomic size difference. This results in the enhancement of the stability of both I- and Z-clusters, which would bring higher glass-forming ability to the alloy system.

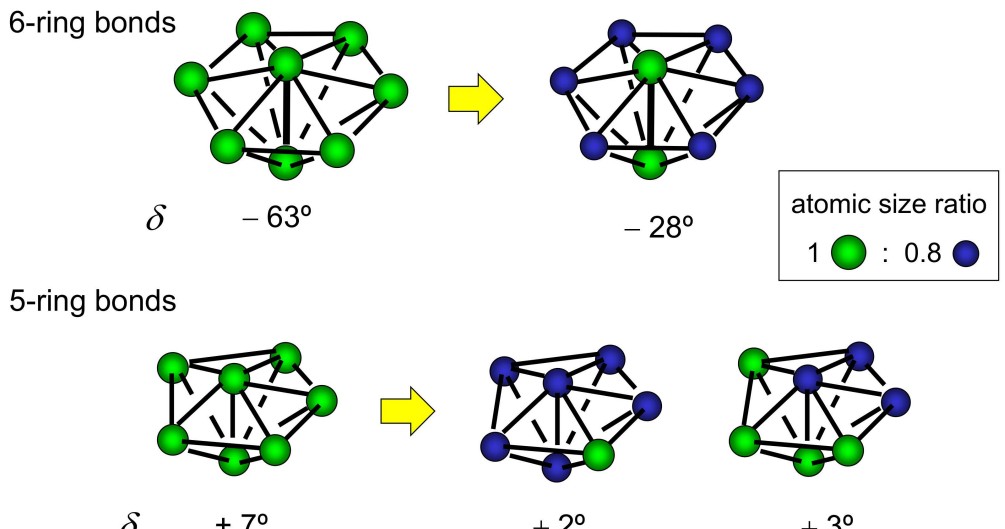

**Figure 13.** Decrease in the deficit angles by introducing the atomic size difference of 0.2 for the 6-ring bonds (**top**) and the 5-ring bonds (**bottom**).

*4.3. Disclination Theory*

In 1983 [24], Nelson applied the idea of the Regge calculus to the physics of liquids and glasses. In his theory, Nelson focused on the sequence of the 4- or 6-ring bonds and called the positive or negative 'disclination' line, respectively. Assuming that the 5-ring bonds have approximately no frustration, the frustration energy is concentrated on these disclination lines. In liquid phases, both positive and negative disclination lines exist and are dynamically moving. On the other hand, in solidified phases, only negative disclination lines remain, because the positive disclination ($\delta = 78°$) has higher frustration than the negative disclination ($\delta = -63°$) and its value has the same sign as that of background 5-ring bonds ($\delta = 7°$), as shown in Figure 12. The point of his theory is in the topology of the network structure formed by the disclination lines in solidified phases, that is, a random network is formed in glassy phases, while an ordered network is formed in the crystalline phases such as the Frank–Kasper phases. This idea is schematically shown in Figure 14.

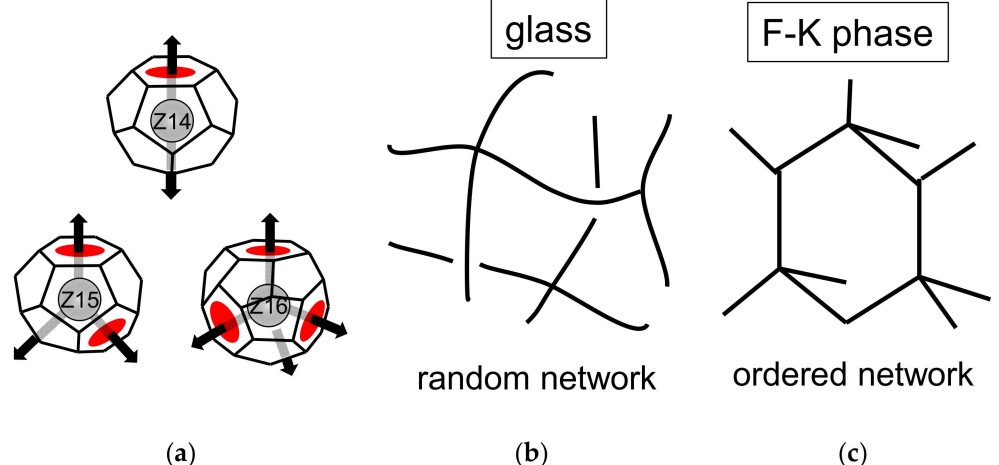

**Figure 14.** (**a**) 6-ring bonds forming a part of disclination lines through Z-clusters. (**b**) Disclination lines in a glassy phase. (**c**) Disclination lines in a Frank–Kasper phase.

In Nelson's disclination theory, the average number $q$ of tetrahedra around each bond is an important order parameter. It is impossible to fill up the whole flat 3-dimensional

space by 5-ring bonds only, because the 5-ring bonds have a small but positive frustration to 7°. To compensate the positive frustration due to 5-ring bonds and keep the average flatness of the whole space, a small amount of negative disclination or 6-ring bonds are needed. Frank has estimated the ideal value $q_{ideal}$ of the lowest frustration state as 5.1043 by embedding the three dimensional configuration into the 4-dimensional space and proposed that an "ideal glass" should be $q = q_{ideal} = 5.1043$. Roughly speaking, because the ratio of the deficit angles of the 6-ring bond to the 5-ring bond is 63/7 = 9, the ratio of the population of the 6-ring bonds to that of the 5-ring bonds should be 1/9 in the "ideal" state, which results in the ideal average number of $q_{ideal} = 5.10$. Note that the Frank–Kasper phases, such as A15, C14, and C14, show the values within $q = 5.10$–5.11 [23,24].

### 4.4. Disclination Lines in Z-Clusters' Network

Following Nelson's idea, we established the 6-ring bond connection or the hexagonal bicap-sharing connection between Z-clusters in the glassy phases in the A-B model alloy system. Figure 15 shows the network structure made of the 6-ring bonds between Z-clusters found in an $r^{BB} = 0.8$ A$_{35}$B$_{65}$ glassy phase formed by fast-cooling (Figure 15a) and in an $r^{BB} = 0.8$ A$_{35}$B$_{65}$ glassy phase formed by slow-cooling (Figure 15b). Comparing to rather small and scattered networks found in the fast-cooling case, the networks have grown in the slow-cooling case. We calculated the cooling rate dependence of the maximum size or the maximum number of directly connected Z-clusters by 6-ring bonds in the $r^{BB} = 0.8$ A$_{1-x}$B$_{x}$ glassy phases. The results are shown in Figure 15c. For all concentration $x$ ranging from 0.5 to 0.7, the maximum size of the Z-clusters' network increases as the structural relaxation takes place.

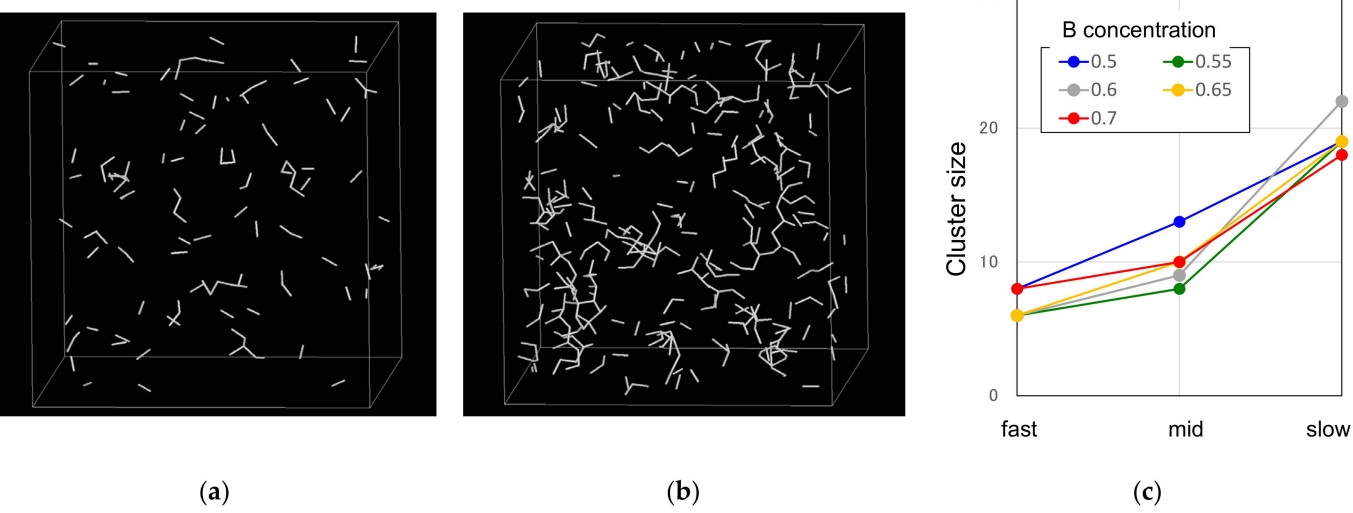

(**a**)　　　　　　　　　　(**b**)　　　　　　　　　　(**c**)

**Figure 15.** (**a**) Network structure of the 6-ring bonds between Z-clusters found in an $r^{BB} = 0.8$ A$_{35}$B$_{65}$ glassy phase formed by fast-cooling and (**b**) in an $r^{BB} = 0.8$ A$_{35}$B$_{65}$ glassy phase formed by slow-cooling. (**c**) Cooling rate dependence of the maximum number of connected Z-clusters by hexagonal bicap sharing in the $r^{BB} = 0.8$ A$_{1-x}$B$_{x}$ glassy phases.

### 4.5. Role of I-Clusters in Z-Clusters' Network

In the next stage, we try to clarify the structural role of I-clusters in the Z-clusters' disclination network. Ding et al. has pointed out in their simulation study [28] that there is a linear correlation between the populations of I- and Z16 cluster in the Cu$_{64}$Zr$_{36}$ supercooled liquid phases. In Figure 16a, we plotted the correlation between the populations of I- and Z-clusters for fifteen $r^{BB} = 0.8$ A$_{1-x}$B$_{x}$ glassy phases formed by different cooling rates with concentration $x$ ranging from 0.5 to 0.7. We can also find an approximately linear correlation between the two. It suggests that the structural relaxation and the growth of the icosahedral network would evolve in a hand-in-hand way for both clusters. Focusing on the hexagonal structural unit shown in Figure 10b, the ratio between the populations

of I- and Z-clusters is 6/2 = 3.0. By considering the I/Z ratio as a new-order parameter to characterize the structural property, we calculated the I/Z ratio for the fifteen glassy phases shown in Figure 16a. The results are shown in Figure 16b. A trend can be found whereby the I/Z ratio slightly decreases as the structural relaxation takes place. However, the relaxed values of the I/Z ratio are around 4.0–4.4 and much higher than the 3.0 value of the hexagonal unit. This indicates that it is difficult to understand the overall structure of the icosahedral network simply by connecting the hexagonal units.

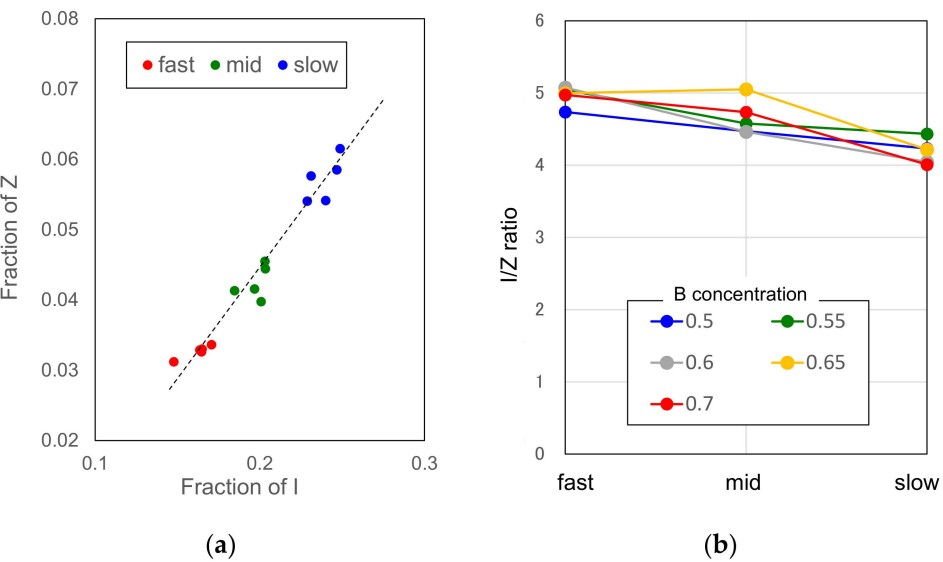

|     |     |
| :-: | :-: |
| (**a**) | (**b**) |

**Figure 16.** (**a**) The correlation between the populations of I- and Z-clusters for fifteen $r^{BB} = 0.8\,A_{1-x}B_x$ glassy phases formed by different cooling rates for $x = 0.5$–0.7. (**b**) The cooling rate dependence of the I/Z ratio of the $r^{BB} = 0.8\,A_{1-x}B_x$ glassy phases for $x = 0.5$–0.7.

To explore the topological character of I-clusters around the Z-clusters' disclination line at the atomic level, we established a 6-ring bond sharing network formed by eleven connected Z-clusters (three Z14s, two Z15s, and six Z16s) found in a glassy phase of the $r^{BB} = 0.8\,A_{35}B_{65}$ system shown in Figure 15b. Figure 17a shows a 'bone' structure formed by central atoms of eleven Z clusters connected by 6-ring bonds. Within the nearest neighbors of the 11 atoms centered at Z-clusters, 33 atoms centered at I-clusters are included (the I/Z ratio is 3.0) and mutually connected by 5-ring bonds to form an I-cluster network wrapping the Z-bone network, as shown in Figure 17b. The surrounding I-cluster network includes five hexagonal ring units (the inset of Figure 9b), among which three rings are penetrated by a 6-ring bond of Z-clusters, as shown in Figure 10b. As shown in Figure 17c, the network formed by connected eleven Z-clusters includes 90 atoms in total. For all bonds shown in Figure 17c, the average number $q$ of tetrahedra around each bond is calculated as 5.08, which is near the value $q_{ideal} = 5.10$ of "ideal glass". It indicates that the Z-clusters' random network wrapped by I-clusters would be a good candidate for a model of "ideal glass" or, maybe, of metallic glasses. On the other hand, for all bonds included in the whole system in the $A_{35}B_{65}$ glassy alloy, $q$ is calculated as 4.99. It means that there is still a lot of free volume, which might include many 4-ring bonds, and that the phase is far from the "ideal glass" in total.

### 4.6. CRN Model for Z-Disclination Network

The structural feature of the disclination model, i.e., the same topology in short range but different topologies in medium range between glass and crystal, reminds us of the continuous random network (CRN) model [36] for amorphous carbon. In the crystalline phases of carbon as graphite or diamond, the network of sp2-type bonds or sp3-type bonds, respectively, has an ordered structure in a long-range scale, while in amorphous

phases, a random network is formed by the same short-range bonding. Since there are two different types of short-range order, sp2 and sp3, the property of amorphous phases is different depending on the ratio of sp2/sp3 bonds. If the sp3 bonds are dominated in an amorphous phase, it is called tetrahedral amorphous carbon (ta-C), as schematically shown in Figure 18a. In a similar manner, as a model for metallic glasses, we can suggest a random network model of Z-clusters connecting by 6-ring bonds, in which three different types of short-range order exist, as schematically shown in Figure 18b. In addition, around the 'bone' structure made by Z-clusters' random network, another type of network formed by I-clusters would always be surrounding in a manner, as shown in Figure 17b. Consequently, the total network formed by I- and Z-clusters will cover the almost whole space in glassy phases just as shown Figure 8b.

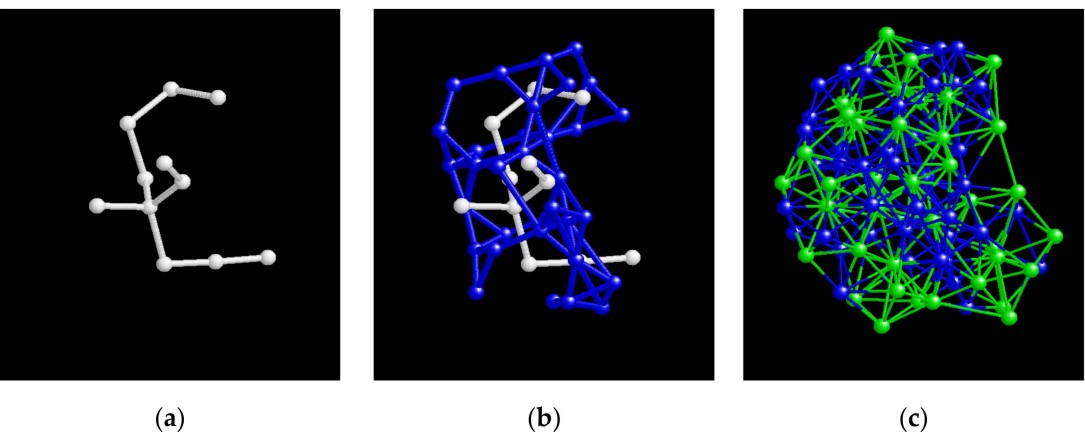

**(a)**                                     **(b)**                                     **(c)**

**Figure 17.** Snapshots of a 6-ring bond sharing network formed by eleven connecting Z-clusters found in a glassy phase of the $r^{BB} = 0.8$ $A_{35}B_{65}$ system: (**a**) Z-cluster network linked by 6-ring bonds, (**b**) I-cluster network (blue) linked by 5-ring bonds included as nearest neighbors to the eleven Z-clusters, and (**c**) all atoms belonging the Z-clusters' network, where green and blue spheres denote the A and B atoms, respectively.

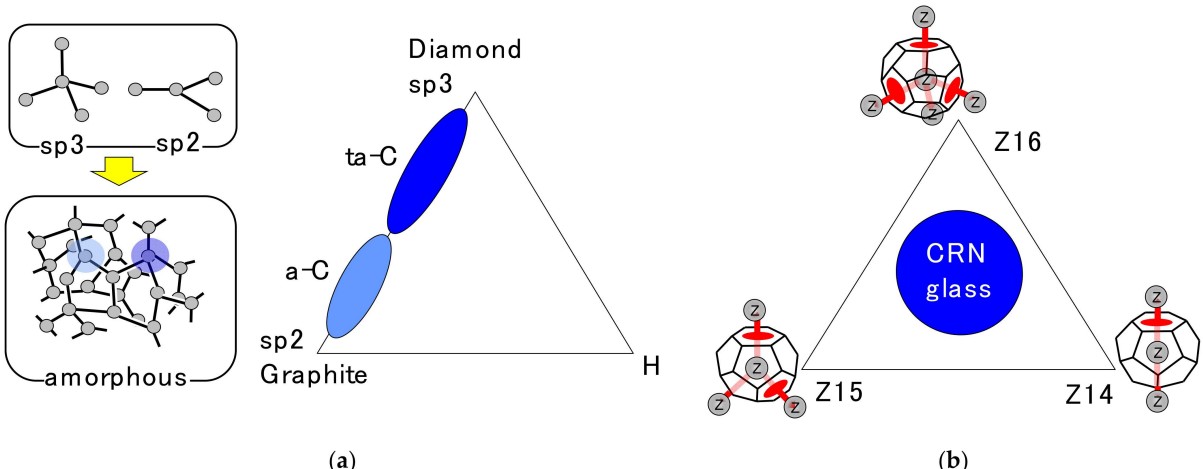

**(a)**                                                     **(b)**

**Figure 18.** (**a**) Short-range order (**left**, **top**) and medium-range order (**left**, **bottom**) in amorphous carbon and a schematic diagram for amorphous carbon (**right**). (**b**) Schematic diagram for CRN glass formed by Z-clusters connected by 6-ring bonds.

Note that we never insist that the Z-clusters' network is main and the I-cluster's one is secondary, but rather that both structures equally contribute to form the whole network in glassy phases and would play complimentary roles to each other. In that sense, I- and Z-clusters have a sort of duality in making a topology of the icosahedral medium-range order. On this point, a deeper understanding of the interrelation between I- and Z-network, such as the structural unit shown in Figure 10b, is needed and it is one of our future tasks.

## 5. Conclusions

The atomic structure of glassy phases in a model A-B binary alloy system, in which the atoms interact with the Lennard–Jones-type potential, is investigated by using molecular dynamics technique. The Frank–Kasper clusters, known as Z14, Z15, and Z16 (Z-clusters), as well as icosahedral clusters (I-clusters), are formed in supercooled liquid phases and their population rapidly increase near the glass transition. The atomic mobility of both I- and Z-clusters is highly suppressed in supercooled liquids, which induces the structural freezing at glass transition. In glassy phases, the population of both I- and Z-clusters increases as the structural relaxation takes place. A considerable atomic size difference between alloying elements would enhance the population of I- and Z-clusters in glassy phases, which would also enhance the glass-forming ability of the alloying system. I- and Z-clusters form a complicated network, which covers the whole space of glassy phases. The main medium-range order between I- and Z-clusters are the connections by 5-ring bonds or 6-ring bonds. By focusing on the 6-ring bond connection between Z-clusters, the basic structure of the glassy phases is understood as a random "disclination" network formed by Z-clusters, which is surrounded by another type of network formed by I-clusters.

**Author Contributions:** Conceptualization, M.S. and H.O.; calculation and investigation, M.S. and H.O.; writing—original draft preparation, M.S.; writing—review and editing, H.O. All authors have read and agreed to the published version of the manuscript.

**Funding:** This research received no external funding.

**Institutional Review Board Statement:** Not applicable.

**Informed Consent Statement:** Not applicable.

**Data Availability Statement:** The data presented in this study are available on request from the corresponding author.

**Conflicts of Interest:** The authors declare no conflict of interest.

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
