# Peer review of "Dual Cluster Model for Medium-Range Order in Metallic Glasses"

_metals, doi:10.3390/met11111840_

Round 1

Reviewer 1 Report

The Author described studies concerning on dual cluster model for medium range order in metallic glasses. The manuscript could be published in Metals after major revision.  Below, several aspects  have mentioned, which should be corrected and some doubts should be explained.

  1. The Abstract should be modified. The most important results should be added in this section.
  2. The motivation of studies should be highlighted in the text.
  3. All magnitudes presented in equations should be explained in the text.

Generally, the Authors did some work. I recommend major revision.

Author Response

Thank you very much for your valuable comments and suggestions.
Our responses to your questions are listed as follows:

Question1: The Abstract should be modified. The most important results should be added in this section.

Response: Thank you for your valuable advice. We have added the sentences that mention important calculation results in the latter part of Abstract.

Question2: The motivation of studies should be highlighted in the text.

Response: Thank you for your thoughtful advice. We have made some correction in the middle part of the last paragraph of Introduction and highlighted the purpose of our study as "The aim of our study is to clarify the topological feature of the icosahedral medium-range order in metallic glasses from the atomistic point of view".

Question3: All magnitudes presented in equations should be explained in the text.

Response: Thank you for your careful reading and comment. We have added sentences to explain the parameters and variables used in the equations more carefully, and to describe the units used in the paper more specifically in Section 2.1.

Reviewer 2 Report

This article builds a Dual Cluster Model for Medium-Range Order in Metallic 2

Glasses,which is interesting for the readers in MD and metallic glasses fields. After answering the following questions, it can be published in Metals:

  1. The Z14, Z15, and Z16 clusters has two, three, and four hexagonal Voronoi faces, respectively. However, only one, two and three hexagonal Voronoi faces for Z14, Z15, and Z16 clusters are marked by red in Fig. 1.  Please give the illustration in the text.
  2. In Fig.2a, it can be seen that the temperature dependence of potential energy is slightly deviated from the linear correlation above Tg. It might be better to use the temperature dependence of volume to determine Tg because of the good linear correlation between volume and temperature in some systems.
  3. In Fig. 3, the population and MSD of different Z-clusters (Z14, Z15, Z16) should also be presented. The stability of these three Z-clusters might be quite different.
  4. Fig. 7, authors show “Possible linking patterns between I- or Z-clusters (insets) and the dependence of their population in the A50B50 glassy phase on the cooling rate”. is it possible choose one rate and give the similar population during the cooling at different different T, and compare them with the earlier work in reference (Phys. Rev. B, 84, 092201 (2011))?
  5. The quality of some figures should be improved.

Author Response

Thank you very much for your valuable comments and suggestions.
Our responses to your questions are listed as follows:

Question1: The Z14, Z15, and Z16 clusters has two, three, and four hexagonal Voronoi faces, respectively. However, only one, two and three hexagonal Voronoi faces for Z14, Z15, and Z16 clusters are marked by red in Fig. 1.  Please give the illustration in the text.

Response: Thank you for your careful reading and advice. We have added the pale red marks onto the hexagonal Voronoi faces on the blind sides of Voronoi polyhedra in Figures 1 and 11.

Question2: In Fig.2a, it can be seen that the temperature dependence of potential energy is slightly deviated from the linear correlation above Tg. It might be better to use the temperature dependence of volume to determine Tg because of the good linear correlation between volume and temperature in some systems.

Response: Thank you for your valuable suggestion. We have replaced Figure 2b from the temperature dependence of atomic energy to the temperature dependence of atomic volume. As you pointed out, the linear behavior of the plotted data has become better.

Question3: In Fig. 3, the population and MSD of different Z-clusters (Z14, Z15, Z16) should also be presented. The stability of these three Z-clusters might be quite different.

Response: Thank you for your valuable comment. We have shown the each population of events using calculation of MSD for Z14, Z15, and Z16 (5, 17, and 18, respectively) in the text, together with a comment on the relative stability between Z14, Z15, and Z16 in supercooled liquids.

Question4: Fig. 7, authors show “Possible linking patterns between I- or Z-clusters (insets) and the dependence of their population in the A50B50 glassy phase on the cooling rate”. is it possible choose one rate and give the similar population during the cooling at different T, and compare them with the earlier work in reference (Phys. Rev. B, 84, 092201 (2011))?

Response: Thank you for your enlightening suggestion and letting us know about the paper Phys. Rev. B, 84, 092201 (2011). We have added this paper newly as Reference [32] and performed similar calculations for the model A-B alloy system. For a fixed cooling rate, we have calculated the temperature dependence of the population of four linking patterns in a cooling process of the A50B50 alloy system and the results are added as Figure 7d. In contrast to the results by Pan et al. in the case of amorphous iron formation, the ratios between four different type connections are almost unchanged during cooling for the model alloy system.

Question5: The quality of some figures should be improved.

Response: Thank you for your valuable advice. The quality of some figures might have got worse in the process of pasting and reducing them into the Word file. So I will send the original figure files separately to Editors.

Reviewer 3 Report

The introduction provides sufficient background and includes all relevant references. The results are clearly presented and the conclusions are supported by the results . I recommend the publishing of the paper as it is now.

Author Response

Thank you for your encouraging comment. Following the suggestions from other reviewers, we have newly calculated the temperature dependence of the population of four different type connections between clusters in a cooling process for the A50B50 alloy system and added a new Figure 7d, in which the calculation results are shown.

Round 2

Reviewer 1 Report

The Authors took into account all reviewers comments and improved the manuscript. It could be published in Metals